# Stable antivortices in multiferroic $\varepsilon$-Fe$_2$O$_3$ with the coalescence of misaligned grains

**Wuhong Xue** [1,6] ✉, **Tao Wang**[1,6], **Huali Yang**[2,6], **Huanhuan Zhang**[3], **Guohong Dai** [4], **Sheng Zhang**[1], **Ruilong Yang**[1], **Zhiyong Quan**[1], **Run-Wei Li** [2], **Jin Tang** [3] ✉, **Cheng Song** [5] ✉ & **Xiaohong Xu**[1] ✉

Antivortices have potential applications in future nano-functional devices, yet the formation of isolated antivortices traditionally requires nanoscale dimensions and near-zero magnetocrystalline anisotropy, limiting their broader application. Here, we propose an approach to forming antivortices in multiferroic $\varepsilon$-Fe$_2$O$_3$ with the coalescence of misaligned grains. By leveraging misaligned crystal domains, the large magnetocrystalline anisotropy energy is counterbalanced, thereby stabilizing the ground state of the antivortex. This method overcomes the traditional difficulty of observing isolated antivortices in micron-sized samples. Stable isolated antivortices were observed in truncated triangular multiferroic $\varepsilon$-Fe$_2$O$_3$ polycrystals ranging from 2.9 to 16.7 $\mu$m. Furthermore, the unpredictability of the polarity of the core was utilized as a source of entropy for designing physically unclonable functions. Our findings expand the range of antivortex materials into the multiferroic perovskite oxides and provide a potential opportunity for ferroelectric polarization control of antivortices.

Vortex and antivortex are a twin of topological solitons that widely exist in nature, such as black holes, tornadoes, screw dislocations in crystals, etc[1,2]. The alignment of spins can also form nontrivial topological magnetic vortices or antivortices that can be used as spintronic functional devices[3]. Interesting physical properties in magnetic vortex states have been disclosed, such as gyroscopic rotation[4,5], polarity reversal[6,7], core motion[8], and vortex pair excitation[9]. These fascinating properties attracted extensive attention for next-generation magnetic data storage devices and other spintronic applications[10–14]. Compared to magnetic vortices, the research progress on magnetic antivortices has been very slow because they are rarely observed due to a magnetic antivortex inevitably generating stray fields causing relatively higher energies. Moreover, large magnetocrystalline anisotropy (MCA) and lateral size can also hinder the stability of this noncollinear topological magnetic structure (Supplementary Fig. 1). Antivortices were first observed between a pair of vortices with the same chirality, forming a vortex-antivortex-vortex configuration[15–18], as listed in Table 1. Unfortunately, this configuration is unstable and easily collapses into a single vortex state. Later, isolated magnetic antivortices were obtained in some deliberately designed materials with specific geometric shapes including '∞'[19–21], '#'[22–24], 'φ'[25,26], and '×'[27,28]. These cases are limited to permalloy with near-zero MCA and nanoscale dimensions. In addition, the lithography process and the specific magnetic field sequence treatments required to generate these isolated antivortices are inconvenient for practical application. Even worse, the antivortex core is difficult to be observed directly in experiments and its application demonstration remains elusive.

[1]Key Laboratory of Magnetic Molecules and Magnetic Information Materials of Ministry of Education & School of Chemistry and Materials Science of Shanxi Normal University, Taiyuan, China. [2]CAS Key Laboratory of Magnetic Materials and Devices, Ningbo Institute of Materials Technology and Engineering, Chinese Academy of Sciences, Ningbo, China. [3]School of Physics and Optoelectronic Engineering, Anhui University, Hefei, China. [4]School of Physics and Materials Science & Institute of Space Science and Technology, Nanchang University, Nanchang, China. [5]School of Materials Science and Engineering, Tsinghua University, Beijing, China. [6]These authors contributed equally: Wuhong Xue, Tao Wang, Huali Yang. ✉e-mail: xuewuhong@sxnu.edu.cn; jintang@ahu.edu.cn; songcheng@mail.tsinghua.edu.cn; xuxh@sxnu.edu.cn

**Table 1 | Comparison of our work with experimental reports on the elements that generate antivortices over the past decades including material system, shape, magnetocrystalline anisotropy constant, lateral size, etc**

| Material | Shape | Anisotropy constant | Lateral size | Isolated antivortex | Applied magnetic field | Ref. |
|---|---|---|---|---|---|---|
| Permalloy | | 0 | 50–200 nm | NO | NO | [15–18] |
| Permalloy | | 0 | <1 µm | YES | YES | [19–28] |
| 2D $\varepsilon$-Fe$_2$O$_3$ | | $2 \times 10^5$ J/m$^3$ | 2–16 µm | YES | NO | This work |

The application requirements of miniaturization, integration, and low power consumption of post-Moore spintronic devices are driving research into topological magnetic antivortex textures in two-dimensional (2D) magnets. However, in 2D magnets, the MCA is essential for stabilizing long-range ferromagnetic order against thermal disturbances[29], which makes the realization of antivortices in 2D magnets particularly challenging. Overcoming the limitations imposed by MCA and size to generate topological magnetic antivortices is crucial for developing compact and low-energy spintronic devices.

Chemical vapor deposition (CVD) is a powerful technique for preparing high-quality 2D materials, offering unique advantages in controlling their size, thickness, and shape[30,31]. Considering the symmetry relationship between the epitaxial layer and substrate, natural 2D polycrystals with "cross-like" grain boundaries can be obtained on demand (Supplementary Fig. 2)[32,33]. Motivated by these considerations, we successfully fabricated truncated triangular 2D $\varepsilon$-Fe$_2$O$_3$ polycrystals through the coalescence of misaligned grains using CVD. We observed stable isolated magnetic antivortices in $\varepsilon$-Fe$_2$O$_3$ polycrystalline nanosheets with a large MCA constant ($K_{MC}$) of ~$2 \times 10^5$ J/m$^3$ and a large lateral size range of 2.9–16.7 µm, without applying a magnetic field. The center of $\varepsilon$-Fe$_2$O$_3$ polycrystals features a clear core composed of an out-of-plane magnetization component with a polarity of $P = +1$ (upward) or $P = -1$ (downward). The magnetization around the core alternately points toward and away from it. Rather than hindering the formation of antivortices, the large MCA value plays a crucial role in their stabilization. This work presents an innovative approach to overcoming the limitations imposed by MCA and size on the generation of magnetic antivortices, providing a guideline for creating antivortices in 2D polycrystals.

## Results

$\varepsilon$-Fe$_2$O$_3$ with a large MCA (~$2 \times 10^5$ J/m$^3$) was selected to create a magnetic antivortex, which possesses an orthorhombic crystal structure with three octahedral Fe sites (Fe$_A$, Fe$_B$, Fe$_C$) and one tetrahedral Fe site (Fe$_D$) (Fig. 1a). The four Fe$^{3+}$ ions exhibit uncompensated antiferromagnetic coupling along the crystallographic $a$-axis and generate net magnetization in this direction[34–36]. The 2D non-layered $\varepsilon$-Fe$_2$O$_3$ was synthesized using a space-confined CVD method (Supplementary Fig. 3). As shown in Fig. 1b, the resulting nanosheets have truncated triangle morphology due to the high symmetry directions of 2D material aligning with the high symmetry directions of the substrate during the CVD process. On a six-fold symmetric mica substrate, the twofold symmetric $\varepsilon$-Fe$_2$O$_3$ grains preferentially grow at three equivalent low-energy orientations with anti-clockwise orientation angles of 0°, 60°, and 120° (Supplementary Fig. 4). These misaligned grains nucleate within a limited range. As the edges of these grains come into contact, three flat grain boundaries are formed. The uniform chemical environment within a small area of the substrate ensures

nearly equal growth rates for the three grains, resulting in a polycrystalline island with a truncated triangular shape and three-grain boundaries at 120° angles to each other. X-ray photoelectron spectroscopy (XPS) and Raman spectra confirm that the synthesized nanosheets are $\varepsilon$-Fe$_2$O$_3$ (Supplementary Fig. 5). The composition ratio of Fe$^{3+}$ in octahedral sites to that in tetrahedral sites is close to 3 confirmed through the deconvoluted XPS Fe 2p$_{3/2}$ peak and atomic-resolution scanning transmission electron microscopy (STEM) (Supplementary Figs. 6, 7 and Table 1). The grain boundaries formed by the coalescence of misaligned grains in 2D $\varepsilon$-Fe$_2$O$_3$ polycrystals have high quality with no current leakage at the grain boundary and ferroelectricity comparable to that of single crystals[36]. Electron backscatter diffraction (EBSD) was used to study the symmetry and orientation of the grains. Figure 1c shows a phase map of the truncated triangular nanosheet with a uniform color distribution, indicating the formation of pure $\varepsilon$-phase Fe$_2$O$_3$. In the inverse pole figure (IPF) map along the out-of-plane $Z$ direction as shown in Fig. 1d, the nanosheet displays a red color, encoding the [001] crystal orientation. The IPF along the in-plane $X$ direction reveals three distinct grains spliced to form the grain boundaries at 120° angles (Fig. 1e), highlighting the polycrystalline characteristic of the sample. These grains exhibit different in-plane orientations, including [010], [100], and [110]. Thus, while the three grains grow in the same out-of-plane direction, their in-plane orientations differ. To further confirm the high-quality coalescence of misaligned grains, $\varepsilon$-Fe$_2$O$_3$ nanosheet was transferred to a copper grid for TEM characterization. Figure 1f–i collected the high-resolution images of two edges (regions I and II), one vertex (region III), and the center (region IV) of the truncated triangular nanosheet. Grain boundaries on the two edges were not observed, but a straight grain boundary was found at the top corner area (marked by the white dashed line in Fig. 1h). The measured lattice plane spacing is ~0.25 nm, corresponding to the (200) crystal planes of $\varepsilon$-Fe$_2$O$_3$[37]. This indicates that the crystallographic $a$-axis (the easy magnetization axis) of the grains is perpendicular to the edge of the nanosheets. Although this orientation is not conducive to the formation of magnetic vortices, it can stabilize antivortices. The corresponding selected area electron diffraction (SAED) patterns are shown in the insets of Fig. 1f–h, reflecting the orthorhombic symmetry of the single crystal grains in the [001] zone axis[38]. In the central region of the $\varepsilon$-Fe$_2$O$_3$ nanosheet, three grain boundaries formed by grain merging are also clearly visible (Fig. 1i).

The successful creation of topological magnetic antivortices in truncated triangular $\varepsilon$-Fe$_2$O$_3$ nanosheets was confirmed using magnetic force microscopy (MFM). Figure 2a, b shows the topological morphology and MFM phase image of 2D $\varepsilon$-Fe$_2$O$_3$ with a thickness of ~15.5 nm (Supplementary Figs. 8 and 9), respectively. The surface of the nanosheet is flat with sharp edges, indicating high crystallinity. In the MFM phase image, the dark contrast (purple) and light contrast (yellow) represent magnetic attractive and repulsive interactions,

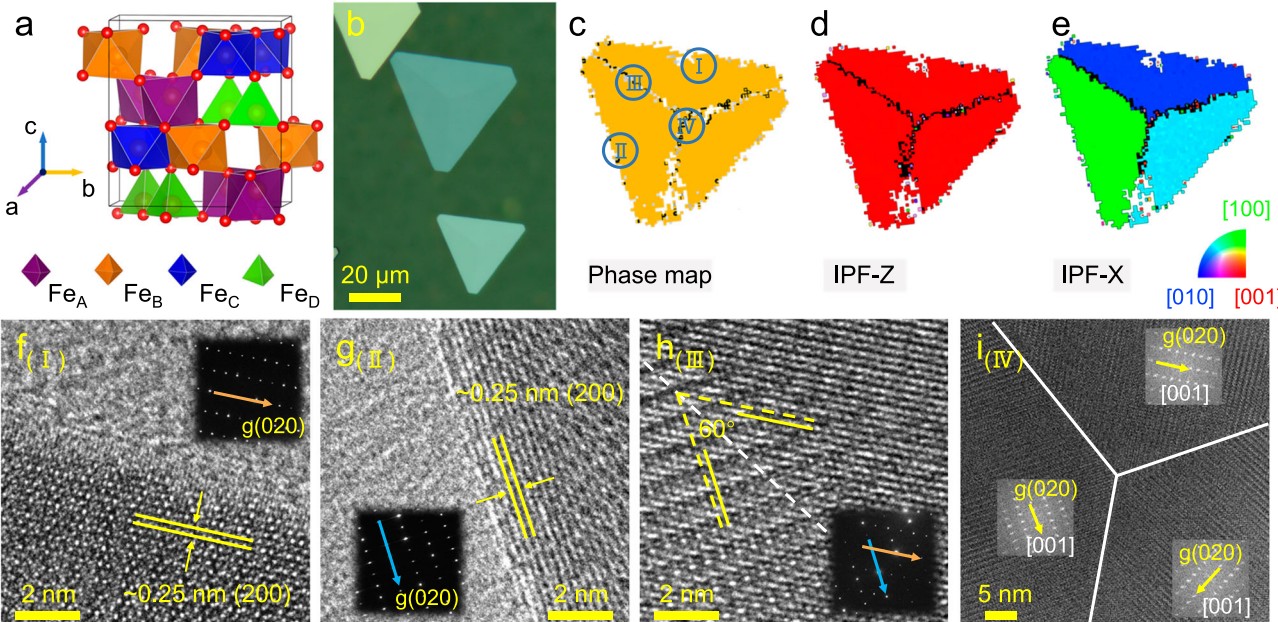

**Fig. 1 | The crystal structure of synthesized 2D ε-Fe₂O₃. a** Crystalline model of ε-Fe₂O₃ unit cell. ε-Fe₂O₃ exhibits an orthorhombic structure, featuring three octahedral sites (Fe$_A$, Fe$_B$, and Fe$_C$) and one tetrahedral site (Fe$_D$). **b** Optical micrograph image of truncated triangular ε-Fe₂O₃ nanosheets on mica. Color-coded phase map (**c**) and inverse pole figure map along the out-of-plane Z direction (IPF-Z) (**d**) and in-plane X direction (IPF-X) (**e**) obtained through electron backscatter diffraction characterization. The color-coded legend is visible on the lower right corner of the map. High-resolution TEM images at the two long edges (**f**, **g**), one vertex (**h**), and center (**i**) of the nanosheet. The corresponding positions are labeled I, II, III, and IV in (**c**). Inset: Corresponding SAED in (**f**–**h**) and fast Fourier transformation pattern in (**i**) along the [001] zone axis of ε-Fe₂O₃. The vector **g** (020) indicates the direction of the crystallographic *b*-axis.

respectively, between the probe and the nanosheet surface. The clear contrasts at the center and edges of the nanosheet confirm strong out-of-plane stray fields. As shown in Fig. 2b, there is a core with magnetization curled towards the out-of-plane direction in the center, and the light and dark contrasts are alternately distributed along the three edges of the nanosheet. Meanwhile, the magnetization, constrained by the MCA, turns towards the in-plane direction in regions away from the center, resulting in a magnetic phase contrast similar to that of the non-magnetic mica substrate[39]. Consequently, the magnetization flows into the core from two opposite directions and sweeps out of the core from the other two directions, conforming to a typical magnetic antivortex configuration. To eliminate potential interference from fluctuations in the nanosheet surface morphology, we reversed the magnetization of the probe[40]. The MFM phase image obtained from repeated testing (Fig. 2c) shows the same reversed contrast, indicating that the contrasts are entirely due to long-range magnetic interactions rather than short-range Coulomb repulsion between the tip and the surface of the nanosheet. Figure 2d depicts the structural model of the truncated triangular ε-Fe₂O₃ nanosheet based on the obtained results of EBSD and TEM characterization. The unit cells with equivalent low-energy orientations are displayed on the corresponding crystal domains, where the crystallographic *a*-axis (easy magnetization axis) is perpendicular to the edges of the nanosheet. Using this structural model and the general parameters of ε-Fe₂O₃, we successfully reproduced the magnetic antivortex structure through micromagnetic simulations. Figure 2e, f illustrates the simulated magnetization alignment and corresponding MFM phase image, respectively. The magnetization is aligned along the easy axis, accumulating magnetic charges at the edges of the nanosheet and generating stray fields. To minimize the magnetic free energy, one of the grains transitions from a single magnetic domain into two domains with opposite magnetization directions along the easy axis. This transition reduces the stray field energy by increasing the domain wall energy. The subtle energy balance contributes to an isolated magnetic antivortex ground state. The above result indicates that the MCA, which is

traditionally believed to be unfavorable for the formation of anti-vortices, plays an indispensable role in stabilizing antivortices in 2D polycrystals. There are two fundamental freedoms, namely polarity $P$ and circulation $c$, which describe the antivortex. Since our samples lack a chiral magnetic interaction (such as the Dzyaloshinskii-Moriya interaction), $P$ and $c$ are not coupled together. Both our experiments and simulations reveal the existence of four types of antivortices, characterized by the combinations $(c, P) = (1, -1), (1, 1), (-1, -1),$ and $(-1, 1)$ with a topological charge $Q = 0.5 P$ (Supplementary Fig. 10). Four styles of antivortices are degenerate with identical total free energy (Supplementary Fig. 11).

In the as-grown truncated triangular ε-Fe₂O₃ nanosheets, we find that magnetic antivortices are not sensitive to the thickness and lateral size of the nanosheets. Within a thickness range of 6.6–42.6 nm and a lateral size range of 2.9–16.7 μm, magnetic antivortices can stably exist (Supplementary Fig. 12). This wide range of dimensions provides significant tolerance and flexibility for the generation and application of magnetic antivortices. Our simulations have identified three potential stable phases—antivortex, vortex, and triple-domain—in truncated triangular ε-Fe₂O₃ nanosheets at zero magnetic field (Supplementary Fig. 13). It is noted that a single-domain state with uniform magnetization (Supplementary Fig. 13d) cannot be stabilized at zero field due to the strong MCA. By evaluating and comparing the total energy of these three phases—antivortex, vortex, and triple-domain—we have constructed a stable phase diagram that varies with the thickness ($t$) and lateral length ($L$) (Supplementary Fig. 14). The antivortex phase exhibits the highest stability across a wide range of geometric parameters. The MCA is crucial in stabilizing the antivortex. Consequently, we also delve into how MCA impacts the stability of the antivortex by simulations. By starting with the antivortex as the initial state, we observe that although the magnetic configurations can change, they remain as antivortex even when the MCA drops to zero (Supplementary Fig. 15). For low MCA values, the antivortex can exist as a metastable phase, possessing a total energy that is higher than that of the vortex (Supplementary Fig. 16).

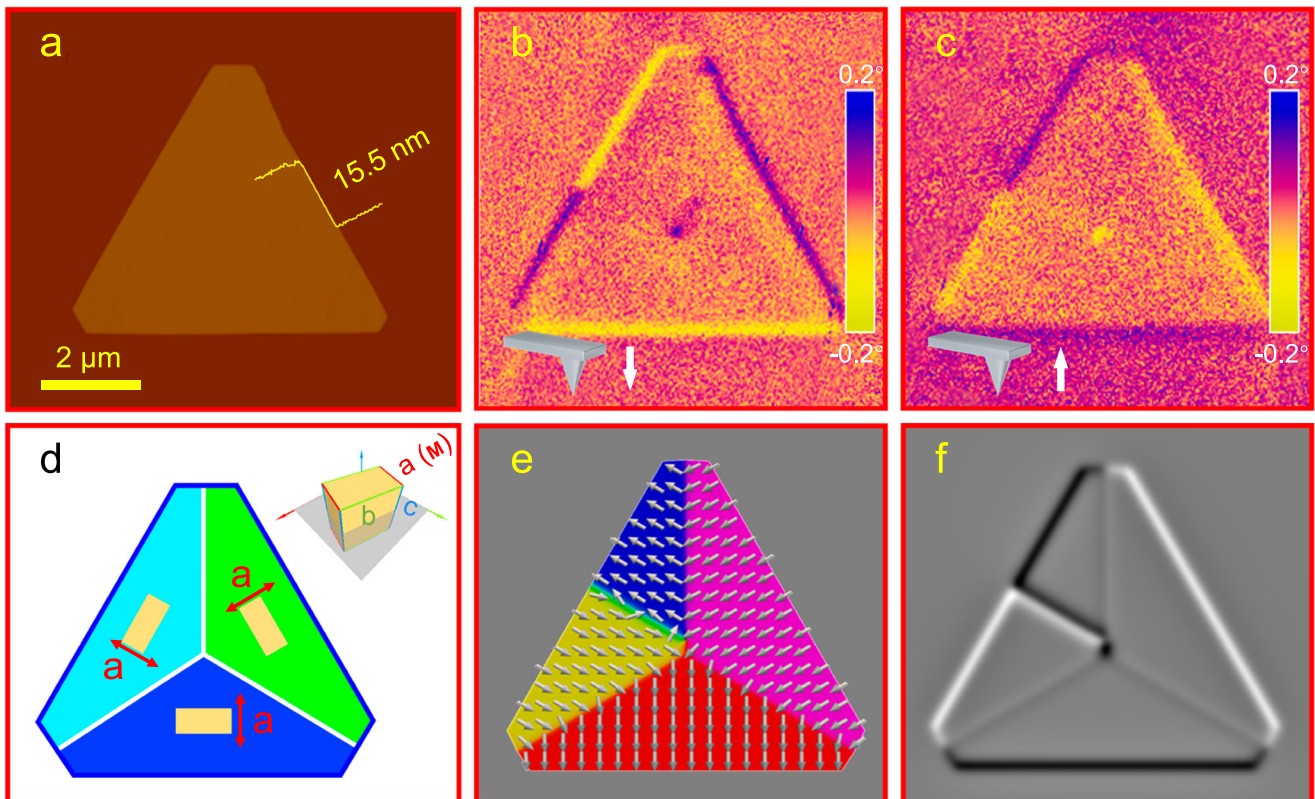

**Fig. 2 | Magnetic antivortex structure of synthesized 2D $\varepsilon$-Fe$_2$O$_3$. a** AFM image of a typical truncated triangular $\varepsilon$-Fe$_2$O$_3$ polycrystal (with a thickness of ~15.5 nm). Corresponding MFM phase images captured with a probe magnetized downward (**b**) and upward (**c**). **d** Schematic diagram of the structure of $\varepsilon$-Fe$_2$O$_3$ polycrystal with truncated triangles. Unit cells with different orientations are covered on the corresponding crystal domains, and the corresponding three-dimensional model is displayed in the upper right corner. The red arrows in the figure indicate the direction of the crystallographic $a$-axis. Micromagnetic simulation of magnetization distribution (**e**) and corresponding simulated MFM phase image (**f**) in $\varepsilon$-Fe$_2$O$_3$ polycrystal.

Isotropic chemical bonds ensure that the crystal structure and magnetic properties of the nanosheets remain consistent despite a decrease in thickness. The large MCA guides the robust magnetization alignment along the easy axis within a single crystal domain. Even if the lateral size of the nanosheets exceeds ten microns, there is no transition from an antivortex to a multidomain state. Furthermore, due to the highly symmetrical geometry of the nanosheet, the magnetization energy of the core is equivalent regardless of whether the magnetization points upwards or downwards. This makes the polarization $P$ (+1 or −1) of the antivortex core randomly assigned, reminiscent of the binary elements '0' and '1'. For example, upward magnetization of the core can be coded as '1' and downward magnetization as '0'. As denoted in Fig. 3a, the polarization of the re-stabilized antivortex after a thermal annealing process should be unpredictable based on the physical mechanism for generating the antivortex (details can be found in the 'method'). To verify this conjecture, continuous thermal annealing was applied to the $\varepsilon$-Fe$_2$O$_3$ nanosheet. As shown in Fig. 3b, the magnetization of the core is initially upward, and a single crystal grain in the lower left corner contains two magnetic domains with opposite magnetization. After the annealing process, the core magnetization switches downward, and the magnetic structure shows two magnetic domains in the upper grain of the truncated triangular nanosheet instead of in the lower left grain as in the initial state. During four annealing processes, the $P$ of the core switches to '0', '1', '0', and '0' states, respectively. This demonstrates that the reconfigured code is indeed randomly generated and unpredictable. We also noticed that any of the three grains with different orientations could divide a magnetic single domain into two domains, which is consistent with the fact that these three grains are equivalent in energy.

The physics-based random assignment of the polarization of the antivortex core is a promising source of entropy for designing Physical Unclonable Functions (PUFs). PUFs can generate unique and unclonable bit-strings by converting the random $P$ distribution of the antivortex core into a random number distribution (Fig. 3c). For example, in a PUF consisting of an array of $\varepsilon$-Fe$_2$O$_3$ nanosheets, a cryptographic key known only to the verifier could be generated by applying a challenge, such as a thermal annealing procedure. This key could be dynamically reconfigured by a simple thermal annealing program, providing new state variables for advanced PUFs design in nanoscale devices. Additionally, we examine the robustness of the magnetic antivortex ground state by applying an external magnetic field and temperature field to the $\varepsilon$-Fe$_2$O$_3$ nanosheets. The results show that the magnetic antivortex state remains stable after applying an external magnetic field of up to 4000 Oe in the out-of-plane direction, or a field of 30 Oe in the in-plane direction (Supplementary Fig. 17). Moreover, the magnetic antivortex state is not disrupted by a high temperature of 500 K, thanks to the high magnetic transition temperature (~500 K) of $\varepsilon$-Fe$_2$O$_3$ (Supplementary Fig. 18)[41,42]. Furthermore, in-situ temperature-variable MFM measurement indicates that the antivortex texture remains stable within the available temperature range (250 K–350 K) that the instrument can achieve (Supplementary Fig. 19). As $\varepsilon$-Fe$_2$O$_3$ is a robust iron oxide under normal temperature and pressure, the antivortex texture is still clear and identifiable after aging for 10 months in ambient condition (Supplementary Fig. 20).

A previous theoretical study demonstrated the emergence and disappearance of new vortices during the stimulation of pulsed in-plane magnetic fields in patterned films[43]. In this work, we delve deeper into the ultrafast magnetic dynamics of initial antivortices in truncated

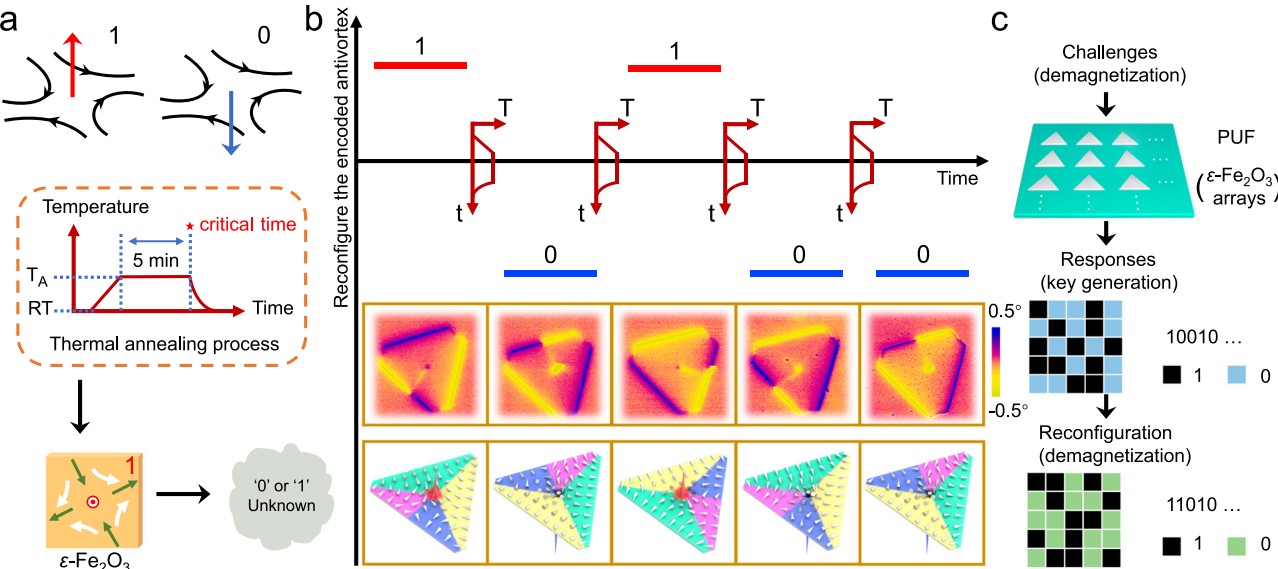

**Fig. 3 | A paradigm of applying magnetic antivortex to PUFs. a** The binary information '1" and '0" can be recorded by the polarity (up and down) of the magnetic antivortex. The initial 2D $\varepsilon$-Fe$_2$O$_3$ is annealed by a thermal annealing process and the polarity of the restabilized antivortex core after is unknown. RT room temperature, $T_A$ annealing temperature. **b** Sketch for reconfiguring encoded antivortex in a 2D $\varepsilon$-Fe$_2$O$_3$ polycrystal. The lower panel displays the corresponding MFM phase images and model of magnetization circulation. **c** Schematic diagram of key generation based on the randomly assigned polarity of antivortex core in a PUFs consisting of an array of $\varepsilon$-Fe$_2$O$_3$ nanosheets.

triangular multiferroic $\varepsilon$-Fe$_2$O$_3$ polycrystals through simulations (Supplementary Fig. 21 and Supplementary movies 1 and 2). We utilize a Gaussian-shaped pulsed in-plane field $\mathbf{B}_{xy}$ along the $y$-axis described by the equation $\mathbf{B}_{xy} = \mathbf{B}_0 \exp(-\frac{(t-0.5ns)^2}{w^2})$, where $\mathbf{B}_0$ represents the peak field, $t$ is time, and $w$ corresponds to the pulse duration. Our simulations reveal a rich array of vortex dynamics, including shifts in the initial core of the antivortex and the creation and annihilation of new antivortices or vortices. These topological defects commonly appear and dissipate at the geometrical edges and grain boundaries. When the pulse width $w$ or peak field $\mathbf{B}_0$ is small, the initial antivortex retains its configuration. However, for stronger or longer pulses, we observe a vortex-antivortex pair as the final configuration. With even stronger and longer pulses, triple-domain states emerge as the stable final configuration.

## Discussion

In summary, our work addresses a crucial aspect of topological magnetic structures by creating isolated magnetic antivortices in polycrystals where MCA is significant. The substrate-induced growth mechanism of 2D polycrystals facilitates the natural formation of stable magnetic antivortices, offering a new approach for the simple and large-scale preparation of magnetic antivortices. Our findings demonstrate the generation of robust magnetic antivortices in 2D $\varepsilon$-Fe$_2$O$_3$ polycrystals with a large MCA ($K_{MC} \approx 2 \times 10^5$ J/m$^3$), overcoming traditional limitations of MCA and lateral size in the generation of antivortices. The distinct antivortex core ensures the feasibility of future manipulation and regulation. This work provides new insights and guidance for expanding the application and understanding of magnetic antivortices.

## Methods

### Preparation and transfer of 2D $\varepsilon$-Fe$_2$O$_3$

2D $\varepsilon$-Fe$_2$O$_3$ was synthesized using a CVD process in a dual-temperature zone tube furnace. Iron chloride tetrahydrate (FeCl$_2$·4H$_2$O) powder and fluorophlogopite mica (KMg$_3$(AlSi$_3$O$_{10}$) F$_2$) were used as the precursor and substrate, respectively. The freshly cleaved mica substrates ($10 \times 10 \times 0.2$ mm) were stacked face-to-face and placed above the precursor to construct a confined synthesis environment. Before growth, the quartz reaction tube was rinsed with 200 sccm of Ar for 20 min. The temperature of the growth area was then raised to 700 °C within 25 min and maintained for 10 min in an O$_2$/Ar atmosphere with a flow rate of 2/198 sccm. Finally, the system was cooled to room temperature naturally under an Ar flow of ~60 sccm.

The as-grown $\varepsilon$-Fe$_2$O$_3$ nanosheets were transferred onto target substrates using polymethyl methacrylate (PMMA) as the transfer medium. Firstly, the PMMA solutions were spin-coated onto $\varepsilon$-Fe$_2$O$_3$/mica and baked at 170 °C for 5 min to improve the adhesion between the nanosheets and PMMA. Next, the PMMA/$\varepsilon$-Fe$_2$O$_3$/mica was immersed in deionized water, and the PMMA film was gently peeled off from the mica substrate using tweezers. Finally, the PMMA membrane floating on the water surface was captured using the target substrates. A mild heating procedure at 70 °C for 30 min was adopted to remove the residual moisture, and then the PMMA was completely removed with acetone.

The detailed steps to refresh the magnetic antivortex through the thermal annealing process are as follows: the as-grown $\varepsilon$-Fe$_2$O$_3$ nanosheets on mica were placed in a furnace and heated. The temperature was raised to 250 °C (above a magnetic transition temperature of $\varepsilon$-Fe$_2$O$_3$) within 8 min, then cooled naturally after 5 min. The entire process was conducted under Ar flow. In the sketch of the thermal annealing process shown in Fig. 3, "$t$", "$T$", "$RT$", and "$T_A$" represent annealing time, furnace temperature, room temperature, and annealing temperature respectively.

### Sample Characterization

Optical microscopy (LW300LJT, CeWei) was used to investigate the morphology of truncated triangular $\varepsilon$-Fe$_2$O$_3$ nanosheets. The symmetry and orientation information of 2D $\varepsilon$-Fe$_2$O$_3$ polycrystals were obtained using a Zeiss Ultra Plus Scanning Electron Microscope (SEM) with an Oxford Electron Backscattering Diffraction system. The chemical composition and vibration modes of $\varepsilon$-Fe$_2$O$_3$ transferred onto Pt/SiO$_2$/Si substrate were detected by XPS (K-Alpha+, ThermoFisher Scientific) and Raman microscopy (LabRAM HR Evolution, laser source wavelength 532 nm, Horiba). The crystal structure, crystalline quality, and elemental

compositions of $\varepsilon$-Fe$_2$O$_3$ nanosheets transferred on the Cu grid were verified by a high-resolution transmission electron microscope (Talos F200X equipped with selected area electron diffraction and TEM energy-dispersive X-ray spectroscopy). Atomic force microscopy (AFM) and MFM characterizations (Dimension Icon, Bruker) were performed on individual $\varepsilon$-Fe$_2$O$_3$ nanosheets to detect the morphology, thickness, and magnetic domain. In MFM measurements, the lift height of the probe coated with magnetic Co/Cr layer (MESP-V2, Bruker) was 120 nm. In-situ temperature-variable MFM measurement was carried out via a Heater/Cooler System compatible with Dimension Icon. The external magnetic field was applied by using a vibrating sample magnetometer (VSM8600, Lake Shore).

## Micromagnetic simulations

The zero-temperature simulations were performed using the GPU-accelerated software- MuMax3[44]. We consider the Hamiltonian including exchange interaction energy, uniaxial magnetic anisotropy energy, Zeeman energy, and dipole-dipole interaction energy. We set the lateral length of the truncated triangular $\varepsilon$-Fe$_2$O$_3$ nanosheets within a range of 0.5 to 8 μm, and the thickness within a range of 3 to 42 nm. We assumed a saturation magnetization $M_S = 118.52$ kA/m, exchange stiffness constant $A_{ex} = 12.0 \times 10^{-12}$ J/m, and damping constant $\alpha = 0.01$. The unit cell (mesh) used in the simulations was 3 nm × 3 nm × 3 nm ($x \times y \times z$). We have divided the entire truncated triangular geometries into three equivalent truncated triangular regions based on experimental results. The easy axis of the uniaxial MCA in each region is oriented perpendicular to the longest edge of the truncated triangular region. When examining the impact of MCA on the stability of vortices and antivortices, we set the MCA constant $K_{MC}$ within a range of 0 to $2 \times 10^5$ J/m$^3$. The equilibrium magnetic state was achieved using the conjugate-gradient method, starting from predefined initial configurations such as antivortex, vortex, triple-domain, and single-domain states. To investigate the effect of pulsed in-plane fields on the stability of an initial antivortex, we obtained dynamic evolutions by advancing the Landau-Lifshitz-Gilbert equations using Runge-Kutta methods. We obtained MFM simulation images with a tip lift height of 50 nm using the tool that has been built in MuMax3.

## Data availability

The dataset of the main figures generated in this study is provided in the Supplementary Information/Source Data file. Source data are provided with this paper.

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

## Acknowledgements

This work was supported by the National Natural Science Foundation of China (Grant nos. U24A6002(X.X.), 12174237(X.X.), 52225106(C.S.), 12241404(C.S.), 12241403(W.X.), 52371245(W.X.), 12474126(H.Y.), 12422403(J.T.) and 12174396(J.T.)), the National Key Research and Development Program of China (Grant nos. 2022YFB3505301(X.X.) and 2024YFA1410200(W.X.)), the Shanxi Province Basic Research Program (Grant no. 202303021224009(W.X.)), and the Shanxi Province Higher Educational Institutions Young Academic Leaders Program (Grant no. 2024Q015(W.X.)).

## Author contributions

W.X., C.S., and X.X. supervised the project. W.X., T.W., and C.S. conceived and designed the experiments. T.W. and S.Z. prepared the materials. W.X. and T.W. performed material characterization, magnetic measurements, and functional demonstration. J.T., H.Y., and G.D. conducted micromagnetic simulation and related discussions. W.X., T.W., H.Y., C.S., and X.X. analyzed the results. W.X., T.W., C.S., and X.X. co-wrote the paper. H.Z., R.Y., Z.Q., R.L., and other authors discussed the results and commented on the manuscript.

## Competing interests

The authors declare no competing interests.
