## [Transparent Peer Review file · Nature Communications]

Stable antivortices in multiferroic ϵ -Fe₂O₃ with the coalescence of misaligned grains

Corresponding Author: Professor Xiaohong Xu

Version 0:

Reviewer comments:

Reviewer #1

(Remarks to the Author)

In this work, Xue et al. proposed a novel approach to forming isolated magnetic antivortices in 2D polycrystals with the coalescence of misaligned grains. By leveraging misaligned crystal domains, the large MCA energy is counterbalanced, which stabilizes the ground state of the antivortex. In addition, the authors innovatively demonstrated the application of antivortex for the physically unclonable functions utilizing the unpredictability of the polarity of the core. This work addresses the limitations imposed by MCA and lateral size on the generation of isolated magnetic antivortex, which is crucial for the development of topological magnetic antivortex based 2D spintronic devices. In my view, this contribution is interesting and deserves to be considered for publication in Nature Communications after a minor revision according to the comments below.

1. The authors claimed that "Stable isolated antivortices were observed in truncated triangular multiferroic ϵ -Fe₂O₃ polycrystals." However, it has been reported that ϵ -Fe₂O₃ prepared by CVD usually exhibits rectangular and hexagonal topographies (Refs. 36, 37). Can antivortex texture occur in these topographies? In addition, how to control the CVD process to generate truncated triangular nanosheets instead of rectangular or hexagonal ones?

2. The authors argued that the coalescence of misaligned grains is decisive for the formation of antivortex texture, but major reported two-dimensional nanosheets prepared by CVD are single crystal. Is this phenomenon reproducible in two-dimensional materials prepared by CVD?

3. In Supplementary Fig. 6, what magnetic structures will the antivortex texture evolve into after applying magnetic fields of 5000 Oe vertically or 40 Oe laterally? Why does a bright contrast appear in the inner region of the nanosheet?

4. Antivortices are unstable compared to vortices, and usually they will move close to nearby vortices and then get annihilated, which is a great challenge for the practical application of antivortices. Can isolated antivortices in ϵ -Fe₂O₃ polycrystals exist for a long time under environmental conditions?

5. Many Figures in the article didn't show a colour bar, such as Figure 3(b) and Supplementary Figs. 5-7.

Reviewer #2

(Remarks to the Author)

The study by Xue et al. reports the method development to receive the isolated antivortex properties with high magnetocrystalline anisotropy, advancing the method of antivortex creation than previously reported. The manuscript is well written, thoughtful and organized. The authors have done several characterization to provide the evidence of antivortices in multiferroic ϵ -Fe₂O₃. However, some of the basic yet crucial experiment are likely ignored. This reviewer is in favor of this manuscript publication in Nature Communication after major revision noted.

1. In addition to in-depth characterization of the sample, I feel something missing is composition ratio of Fe_A, Fe_B and Fe_C at octahedral and tetrahedral site. Authors performed the XPS but only figures are provided in supplementary without obtained data. I suggest to tabulate those obtained data in supplementary or (perform Mossbauer spectroscopy at room temperature to obtain the compositional ratio of iron particles in tetrahedral and octahedral site. It would be great to know their iron

population with hyperfine field of such stable isolated antivortex). Figure 2a, thickness is invisible, probably need HRTEM to visualize close dimension of polycrystals. The detail process and set parameter during AFM experiment and magnetic simulations need to be given in the supplementary. Same for MFM, no information about experimental details and parameter used are unexpected.

2. In page 8, authors have discussed that the antivortices are likely insensitive to the given dimension of samples. The reviewer believes that the vortex and anti-vortex are largely geometry dependent complex structure, thus, I'd encourage authors to provide the phase diagram (Total energy vs size (thickness and/or height) showing region of vortex, antivortex and single domain region of sample. Magnetic simulation is enough. If it is not possible, discuss appropriately somewhere in the main manuscript as it is important message for readers.

3. Figure 3 quality is very poor, especially in Fig 3b (spin-configuration). The reviewer struggled to find the spin direction. Need quality of image clearly showing the antivortex spin direction. In addition, I cannot see the meaning of cross-tie walls as mentioned in conclusion. Can author elaborate this in the manuscript where they want to explain this cross-tie wall? In Fig 3a, I do not understand why the in-plane magnetization (black arrow) directions are same after the polarity changed; if the polarity changed, why not direction? In another schematic figure authors should clearly indicates the specific thermal transition time (for example critical time etc.) when T_c change with time, gets plateau and again back to same. In conclave shape schematic, what are the green and gray arrow representing? May be in-plane magnetization but it is good to clarify the reason of using two different colors.

4. In addition to Supplementary Fig 6, can authors examine how the distortion affect in evolution of such antivortex? I would like to see the distortion of the antivortex structure (may be vortex creation followed by annihilation and form new anti-vortex) induced by an in-plane field? It might help to visualize and predict the new anti-vortices in such materials if distortions play the role in originating new antivortex. It would be great if authors can provide experimentally as well as magnetic simulation. If not experimentally because of limitation, I'm fine with magnetic simulation. For this, paper 10.1103/PhysRevB.77.060404 may be helpful.

5. The reviewer is curious to see the magnetization properties (magnetization vs field and magnetization vs low temperature (5K-350 K). Usually at low temperature magnetocrystalline anisotropy (MCA) changes as compared to room temperature because of geometry change of Fe sites in the crystal structure. I wonder this stable antivortex still exist while changing the MCA.

Reviewer #3

(Remarks to the Author)

Version 1:

Reviewer comments:

Reviewer #1

(Remarks to the Author)

In this revised version, the authors have well addressed my previous concerns. I'm satisfied with their responses and revisions. I'd like to recommend the acceptance.

Reviewer #2

(Remarks to the Author)

I am satisfied with revised versions and authors response and recommend the current version of the manuscript for publication in Nature Communication.

Reviewer #3

(Remarks to the Author)

List of Responses and Revisions

Title: “Stable antivortices in multiferroic ϵ -Fe₂O₃ with the coalescence of misaligned grains”

Authors: Wuhong Xue, Tao Wang, Huali Yang, Huanhuan Zhang, Guohong Dai, Sheng Zhang, Ruilong Yang, Zhiyong Quan, Run-Wei Li, Jin Tang, Cheng Song & Xiaohong Xu

ALL THE CHANGES WERE DONE IN RED IN THE REVISED MANUSCRIPT.

Response to Reviewer 1:

General Comments:

In this work, Xue et al. proposed a novel approach to forming isolated magnetic antivortices in 2D polycrystals with the coalescence of misaligned grains. By leveraging misaligned crystal domains, the large MCA energy is counterbalanced, which stabilizes the ground state of the antivortex. In addition, the authors innovatively demonstrated the application of antivortex for the physically unclonable functions utilizing the unpredictability of the polarity of the core. This work addresses the limitations imposed by MCA and lateral size on the generation of isolated magnetic antivortices, which is crucial for the development of topological magnetic antivortex based 2D spintronic devices. In my view, this contribution is interesting and deserves to be considered for publication in Nature Communications after a minor revision according to the comments below.

Response: We thank the reviewer for his/her positive comments on our work. We have carefully considered the reviewer’s comments/suggestions. According to the reviewer’s valuable suggestions, the manuscript was revised accordingly as in the follow.

Q1: *The authors claimed that “Stable isolated antivortices were observed in truncated triangular multiferroic ϵ -Fe₂O₃ polycrystals.” However, it has been reported that ϵ -Fe₂O₃ prepared by CVD usually exhibits rectangular and hexagonal topographies (Refs. 36, 37). Can antivortex texture occur in these topographies? In addition, how to control the CVD process to generate truncated triangular nanosheets instead of rectangular or hexagonal ones?*

Response: We thank the reviewer very much for this constructive question. In fact, the

formation of antivortices is highly dependent on the geometric shape of the nanosheets. In our previous experiments (*Adv. Mater.* 2024, 36, 2311041), we have also observed rectangular (**Fig. R1a**) and hexagonal samples (**Fig. R1d**). Inverse pole figure maps derived from Electron Back-Scatter Diffraction (EBSD) show that the rectangular nanosheet is single crystal (**Fig. R1c**), while the hexagonal nanosheet is polycrystal composed of several single grains with different orientations (**Fig. R1f**). From the magnetic force microscopy (MFM) results, the rectangular single crystal has a typical magnetic single-domain state, with the magnetization aligned along the easy magnetization axis (crystallographic a-axis) and two opposite magnetic poles appearing on the two long sides of the nanosheet (**Fig. R1b**), and hexagonal polycrystal exhibits the multi-domain configuration, with obvious phase contrast appearing at grain boundaries due to the stray field generated by the magnetic domain walls pinned at the grain boundaries (**Fig. R1e**). This is probably because the large magnetocrystalline anisotropy of ϵ -Fe₂O₃ prevents non-collinear spin textures from appearing in rectangular single crystals, and the geometry of hexagonal polycrystals with more complex grain co-growth and coalescence is difficult to stabilize the magnetic antivortex texture.

The truncated triangular nanosheets can be achieved by lowering the growth temperature during the chemical vapor deposition process. At a growth temperature of $\sim 780^\circ\text{C}$, the ϵ -Fe₂O₃ nanosheets mainly exhibit hexagonal morphology (**Fig. R2a**). When the growth temperature is lowered to $\sim 740^\circ\text{C}$, hexagonal and truncated triangular nanosheets coexist (**Fig. R2b**). At a lower growth temperature of $\sim 700^\circ\text{C}$, the shape of the nanosheets is mainly truncated triangle (**Fig. R2c**). Adjusting the growth temperature can regulate the nucleation density in the chemical vapor deposition process. As shown in **Fig. R3**, with the increase of growth temperature, the saturated vapor pressure of the reactants will be higher. The high concentration of active iron atoms produced by the decomposition of FeCl₂·4H₂O will result in a high nucleation density of ϵ -Fe₂O₃ on mica substrates. Sufficient precursor causes a large number of initial grains to participate in multiple nucleation and then rapidly grow into hexagonal polycrystals. On the contrary, at lower temperatures, the reduced nucleation density and growth rate cause the initial grains to tend to combine in the form of the lowest energy (three equivalent low-energy orientations with misorientation angles of 60° , and 120° between each grain), grow synchronously, and ultimately evolve into truncated triangles.

Figure R1. (a,d) AFM images, (b,e) MFM phase images and (c,f) Color coded inverse pole figures derived from EBSD scan of the rectangular and hexagonal nanosheets.

Figure R2. OM images and the corresponding Raman spectra of hexagonal or (and) truncated triangular ϵ - Fe_2O_3 nanosheets grown at (a) 780°C, (b) 740°C, and (c) 700°C.

Figure R3. Schematic of 2D ϵ -Fe₂O₃ polycrystal shapes changing from truncated triangle to hexagon under the influence of temperature.

Q2: The authors argued that the coalescence of misaligned grains is decisive for the formation of antivortex texture, but major reported two-dimensional nanosheets prepared by CVD are single crystal. Is this phenomenon reproducible in two-dimensional materials prepared by CVD?

Response: We thank the reviewer very much for asking this important question. In most cases, the researchers aim to prepare 2D single crystal to achieve excellent performance. And various strategies such as step edge docking (*Nat. Nanotech.* 2022, 16, 1201), quasi-equilibrium growth (*Nat. Commun.* 2024, 15, 7471), and interisland-distance-mediated growth (*Nano Lett.* 2023, 23, 1758), have been developed and adopted to break the energy degeneracy of equivalent low energy orientations for obtaining 2D single crystals grown with unidirection. However, multiple nucleations are ubiquitous in the chemical vapor deposition processes for 2D materials growth. Under the influence of substrate symmetry, the high symmetry direction of the 2D material is aligned along the high symmetry direction of the substrate. When the symmetric group of the 2D material is a subgroup of the symmetric group of the substrate, multi-orientations of 2D grains naturally occur. For example, threefold symmetric 2D materials have four equivalent low-energy orientations on a fourfold symmetric substrate and fourfold symmetric 2D materials have three equivalent low-energy orientations on a six-fold symmetric substrate (left panel of **Fig. R4**). Multiple 2D grains with different orientations can coalesce to form polycrystals with diverse shapes. The coalescence of four different oriented triangular grains on a fourfold symmetric substrate can form a cross-shaped structure, while the coalescence of three different oriented square grains on a six-fold symmetric substrate can form a truncated

triangular structure (right panel of **Fig. R4**). By selecting appropriate 2D materials and substrates, 2D polycrystals with highly symmetric in geometry can be fabricated. When these 2D materials have appropriate magnetocrystalline anisotropy constants, they can serve as ideal platforms for constructing magnetic antivortex texture.

These data were added in the revised Supplementary Information (**Supplementary Fig. 2**) and the corresponding statement is added in the revised manuscript, see **Page 4**: “*This work presents an innovative approach to overcoming the limitations imposed by MCA and size on the generation of magnetic antivortices, providing a guideline for creating antivortices in 2D polycrystals.*”

Figure R4. Schematic of the equivalent low-energy alignments of threefold (fourfold) symmetric 2D materials on fourfold (six-fold) symmetric substrates, and the principle on the construction of the coalescence of these different oriented triangular or square grains to form geometrically highly symmetric polycrystals.

Q3: *In Supplementary Fig. 6, what magnetic structures will the antivortex texture evolve into after applying magnetic fields of 5000 Oe vertically or 40 Oe laterally? Why does a bright contrast appear in the inner region of the nanosheet?*

Response: We thank the reviewer for posing this important question. The antivortex texture in **Fig. S6** of the initial manuscript (**Supplementary Fig. 17** in the revised Supplementary information) evolves into a triple-domain state composed of three single domains after applying a vertical magnetic field of 5000 Oe or a lateral magnetic field of 40 Oe. **Figs. R5a, R5b** show the MFM phase images and model of corresponding magnetization circulation of these triple-domain states, and the three misaligned crystal domains all exhibit magnetic single-domain states. The magnetization within the magnetic single-domain is aligned along the magnetic easy axis, and the stray fields

generated at the edges of the nanosheet interact with the MFM tip to generate the magnetic-phase signal. These magnetic single domains are separated by magnetic domain walls marked by white and black dashed lines in **Figs. R5a, R5b**. Because the energy of the magnetic domain walls at the grain boundary is greatly reduced, the magnetic domain walls are pinned by the grain boundary. After applying a certain external magnetic field, the antivortex state transforms into a triple-domain state, which is because when an external magnetic field is applied, the Zeeman energy introduced into the system breaks the original energy balance, causing the antivortex state to be altered. After removing the external magnetic field, the remnant magnetic state of 2D ϵ -Fe₂O₃ polycrystals did not return to its original magnetic antivortex ground state but transformed into a metastable triple-domain state with slightly higher energy. The metastable triple-domain state can be restored to antivortex ground state undergoing a thermal demagnetization procedure.

In addition, the bright contrast observed in the inner region of the nanosheets in the MFM phase images is the magnetic signal of the Bloch-type magnetic domain wall. There are three magnetic domain walls in **Fig. R5a, R5b** including Bloch-type domain walls (indicated by black dashed lines) and Néel-type domain walls (indicated by white dashed lines). The magnetization direction of the magnetic tip is along the out-of-plane direction of the nanosheet, so the magnetic tip will only interact with the out-of-plane component of the magnetization inside the nanosheet and generate a magnetic-phase signal. As shown in **Fig. R5c**, the magnetization inside the Bloch-type domain wall has the component along the out-of-plane direction, which is equivalent to generating an out-of-plane magnetic field (B), resulting in an oscillating phase difference between the magnetic tip and the piezoelectric driver during scanning. The magnetization inside the Néel-type domain wall does not have the component along the out-of-plane direction, so it cannot be directly captured by MFM.

Figure R5. The MFM phase images and the corresponding model of magnetization circulation after applying the (a) vertical magnetic field (5000 Oe) and (b) lateral magnetic field (40 Oe). The white and black dashed lines guide the Néel-type domain walls and Bloch-type domain walls, respectively. The scale bar is 2 μm . (c) Scheme of the working mechanism of MFM for detecting magnetic signals of Bloch-type domain wall and Néel-type domain wall.

Q4: *Antivortices are unstable compared to vortices, and usually they will move close to nearby vortices and then get annihilated, which is a great challenge for the practical application of antivortices. Can isolated antivortices in $\epsilon\text{-Fe}_2\text{O}_3$ polycrystals exist for a long time under environmental conditions?*

Response: We thank the reviewer very much for this question. To check the environmental stability of the magnetic antivortex, we exposed a freshly synthesized truncated triangular $\epsilon\text{-Fe}_2\text{O}_3$ polycrystal to ambient condition for 10 months. As shown in **Fig. R6**, the isolated magnetic antivortex is still clear and identifiable, proving the robust environmental stability of the magnetic antivortex texture of 2D $\epsilon\text{-Fe}_2\text{O}_3$ polycrystal. The magnetic antivortex is highly dependent on the geometric structure of $\epsilon\text{-Fe}_2\text{O}_3$ polycrystals, and $\epsilon\text{-Fe}_2\text{O}_3$ structure has an excellent air stability, which provides a guarantee for the stability of the magnetic antivortex.

These data were added in the revised Supplementary information (**Supplementary Fig. 20**) and the corresponding discussion is added in the revised manuscript, see **Page 11**: *“As $\epsilon\text{-Fe}_2\text{O}_3$ is a robust iron oxide under normal temperature and pressure, the antivortex texture is still clear and identifiable after aging for 10 months in ambient condition (Supplementary Fig. 20).”*

Figure R6. MFM phase images of a (a) freshly synthesized and (b) ten-months aged ϵ -Fe₂O₃ nanosheets. Both of them were measured at room temperature. The scale bar is 2 μ m.

Q5: *Many Figures in the article didn't show a colour bar, such as Figure 3(b) and Supplementary Figs. 5-7.*

Response: We appreciate the reviewer's responsible attitude and sincerely apologize for the negligence on the missing colour bars in some Figures in the manuscript. We carefully reviewed the manuscript throughout and the missing colour bar in the figures were added.

Response to Reviewer 2:

General Comments:

The study by Xue et al. reports the method development to receive the isolated antivortex properties with high magnetocrystalline anisotropy, advancing the method of antivortex creation than previously reported. The manuscript is well written, thoughtful and organized. The authors have done several characterization to provide the evidence of antivortices in multiferroic ϵ -Fe₂O₃. However, some of the basic yet crucial experiment are likely ignored. This reviewer is in favor of this manuscript publication in Nature Communication after major revision noted.

Response: We thank the reviewer for his/her precious time on reviewing our manuscript. It is greatly appreciated for the reviewer to provide constructive suggestions, which is quite helpful and significant for us to further improve the quality of the manuscript. According to the reviewer's suggestions, we have conducted some additional experiments and simulations and tried our best to make the revised manuscript more clearly. Our point-to-point response is presented below.

Q1.1: *In addition to in-depth characterization of the sample, I feel something missing is composition ratio of Fe_A, Fe_B and Fe_C at octahedral and tetrahedral site. Authors performed the XPS but only figures are provided in supplementary without obtained data. I suggest to tabulate those obtained data in supplementary or (perform Mössbauer spectroscopy at room temperature to obtain the compositional ratio of iron particles in tetrahedral and octahedral site. It would be great to know their iron population with hyperfine field of such stable isolated antivortex).*

Response: We thank the reviewer very much for this important question. Firstly, we deconvoluted the Fe 2p XPS spectrum to obtain the composition ratio of Fe³⁺ distributed on octahedral (oct) and tetrahedral (tet) sites. Considering the broader, less intense peak structure in the 2p_{1/2} region, as well as the potential for errors in the higher-binding energy regions, the concentration analysis is solely based on the intensity of the Fe main 2p_{3/2} peak. In the Fe2p_{3/2} region, the peak located at 711.1 eV is deconvoluted into two peaks, as shown in **Fig. R1**. The binding energies of the Fe³⁺(oct) and Fe³⁺(tet) peaks are 710.4 eV and 712.5 eV, respectively, which are consistent with the values reported in the literature (*Nanoscale*, 2017, 9, 7858-7867; *J. Magn. Magn. Mater.* 2022, 552, 169218). The fitted yields a Fe³⁺(oct)/Fe³⁺(tet) ratio of 2.98 which is

in reasonable agreement with the expected value of 3 for the ϵ -Fe₂O₃ crystal structure. The related parameters of each component peak, including their positions, relative areas, full width at half maximum (FWHM), and squared deviation (χ^2), are listed in **Table R1**.

Thanks again for the reviewer's professional advice because Mössbauer spectroscopy is a precise probe for accurately detecting the local electronic environment around the Fe ions. It is worth pointing out that the Mössbauer effect requires that the γ -rays must interact with a sufficient amount of iron to produce a high-quality Mössbauer signal, which makes the actual testing process demanding in terms of sample quantity (typically requiring an element density of 10-20 mg/cm²). Unfortunately, the ϵ -Fe₂O₃ nanosheets grown on the substrate surface are thin and discrete, making it difficult to have sufficient Fe content to produce distinct Mössbauer signals. Although we collected the ϵ -Fe₂O₃ nanosheets on multiple substrates by ultrasonic treatment, the absorption peaks were still unidentifiable (**Fig. R2**).

To clarify the composition ratio of Fe³⁺ at the four non-equivalent iron sites, we characterized the transferred samples using atomic-resolution high-angle annular dark-field scanning transmission electron microscopy (HAADF-STEM). As shown in **Fig. R3**, a high-quality lattice structure is observed. The atomic arrangement perfectly matches the lattice structure of ϵ -Fe₂O₃ along the [001] crystal axis. The Fe³⁺ ions located at the Fe_A, Fe_B, Fe_C, and Fe_D sites are represented by spheres with different colors, and it is evident that their ratio among each other is 1:1:1:1. The ratio of octahedral sites to tetrahedral is 3:1, which is consistent with the XPS results.

These data were added in the revised Supplementary information and the corresponding discussion was added in the revised manuscript, see **Page 5**: “*The composition ratio of Fe³⁺ in octahedral sites to that in tetrahedral sites is close to 3 confirmed through the deconvoluted XPS Fe2p_{3/2} peak and atomic-resolution scanning transmission electron microscopy (STEM) (Supplementary Fig. 6,7, and Table 1).*”

Figure R1. Fe2p_{3/2} region of Fe 2p XPS spectrum. Contribution from the Fe³⁺(oct) and Fe³⁺(tet) cations to the Fe2p_{3/2} peak are shown by color-filled deconvoluted peaks.

Table R1. Detailed information on Fe2p_{3/2} component peaks of Fe³⁺ in octahedral (oct) and tetrahedral (tet) sites obtained from the Fe2p_{3/2} region in XPS spectrum of ϵ -Fe₂O₃.

Deconvoluted Region	Peak Position (eV)	Attribution	Relative Area	FWHM (eV)	χ^2	Peak Ratio
Fe2p _{3/2}	710.4	Fe ³⁺ (oct)	2899.7	2.6	1.27	2.98:1
	712.5	Fe ³⁺ (tet)	973.1	2.6		

Figure R2. Mössbauer spectrum of the as-grown ϵ -Fe₂O₃ nanosheets measured at 300 K.

Figure R3. (a) Lattice structures of ϵ -Fe₂O₃ along [100] (side view) and [001] crystal axes (top view). (b) Aberration-corrected STEM image of ϵ -Fe₂O₃ nanosheet.

Q1.2: Figure 2a, thickness is invisible, probably need HRTEM to visualize close dimension of polycrystals.

Response: According to the reviewer's great suggestion, we have added the 3D AFM image of the nanosheet (**Fig. R4**) corresponding to Fig. 2a to the Supporting Information for visualizing the thickness. Meanwhile, we conducted cross-sectional TEM measurement to further determine sample thickness. The high-resolution TEM image and the elemental map reveal that the thickness of the nanosheet is approximately 15.5 nm (**Fig. R5**), which is consistent with the AFM results. The 3D AFM image and TEM image were provided in **Supplementary Figs. 8,9**.

Figure R4. (a) 3D morphology of the ϵ -Fe₂O₃ nanosheet in Figure 2a. (b) Side view of morphology in (a) (upper panel) and the height profile along the white dashed line (lower panel), demonstrating the thickness of the nanosheet is approximately 15.5 nm.

Figure R5. Cross-sectional TEM images with low- and high-magnification of the ϵ -Fe₂O₃ nanosheet in Figure 2a, and the EDS mapping of Fe, showing the uniform distribution of element.

Q1.3: *The detail process and set parameter during AFM experiment and magnetic simulations need to be given in the supplementary. Same for MFM, no information about experimental details and parameter used are unexpected.*

Response: Thank the reviewer for his/her thorough review. We have supplemented the detailed procedures and parameter settings for the AFM, MFM experiments in the Supporting Information (**Text S1**), and micromagnetic simulations in Method section of the revised manuscript (**Pages 14,15**) to ensure the reproducibility and transparency of the research.

Q2: *In page 8, authors have discussed that the antivortices are likely insensitive to the given dimension of samples. The reviewer believes that the vortex and anti-vortex are largely geometry dependent complex structure, thus, I'd encourage authors to provide the phase diagram (Total energy vs size (thickness and/or height) showing region of vortex, antivortex and single domain region of sample. Magnetic simulation is enough. If it is not possible, discuss appropriately somewhere in the main manuscript as it is important message for readers.*

Response: Following the reviewer's nice suggestion, we have added a phase diagram as a function of size by comparing the total energy of vortex (**Fig. R6a**), antivortex (**Fig. R6b**), and triple-domain (**Fig. R6c**) states, as shown in **Fig. R7**. Because of strong magnetocrystalline anisotropy, single domains with uniform magnetizations (**Fig. R6d**) are not stable phases.

Our simulations have demonstrated that the antivortex phase is the most stable across a broad spectrum of material parameters, within a thickness range of 3-39 nm and a lateral size range of 0-9 μm . However, it is worth noting that simulations for larger sizes could not be conducted due to the memory limitations of our GPU. In the low-depth limit, we observed that the total energy of the triple-domain phase is slightly lower than that of the antivortex phase. Additionally, at a height of 39 nm, the total energy of the vortex phase is slightly lower than that of the antivortex phase for triangles with lateral lengths of 1 and 2 μm .

We have incorporated a discussion on the stable phase diagram into the revised manuscript, see **Page 9**: *“Our simulations have identified three potential stable phases—antivortex, vortex, and triple-domain—in truncated triangular $\epsilon\text{-Fe}_2\text{O}_3$ nanosheets at zero magnetic field (Supplementary Fig. 13). It is noted that a single-domain state with uniform magnetization (Supplementary Fig. 13d) cannot be stabilized at zero field due to the strong MCA. By evaluating and comparing the total energy of these three phases—antivortex, vortex, and triple-domain—we have constructed a stable phase diagram that varies with thickness (t) and lateral size (L) (Supplementary Fig. 14). Our simulations further reveal that the antivortex phase exhibits the highest stability across a wide range of geometric parameters”.*

Figure R6. Simulated magnetizations and MFM images of the vortex (a), antivortex (b), triple-domain (c), and single-domain (ferromagnet) (d).

Figure R7. Simulated stable phase diagram based on the thickness and lateral length at zero fields. (a) Simulated stable phase diagram as a function of geometrical size. (b)-(e) Dependence of total energy density on thickness t for a fixed length L at 0.5, 1, 2, and 3 μm .

Q3.1: Figure 3 quality is very poor, especially in Fig 3b (spin-configuration). The reviewer struggled to find the spin direction. Need quality of image clearly showing the antivortex spin direction.

Response: We thank the reviewer for his/her careful reading, and we have improved the quality of **Fig. 3b** to clearly show the antivortex spin direction, as shown in **Fig. R8**.

Figure R8. A paradigm of applying magnetic antivortex to PUFs. a The binary information ‘1’ and ‘0’ can be recorded by the polarity (up and down) of the magnetic antivortex. The initial 2D $\epsilon\text{-Fe}_2\text{O}_3$ is annealed by a thermal annealing process and the polarity of the restabilized antivortex core after is unknown. RT: room temperature. T_A : annealing temperature. b Sketch for reconfiguring encoded antivortex in a 2D $\epsilon\text{-Fe}_2\text{O}_3$ polycrystal. The lower panel displays the corresponding MFM phase images and model of magnetization circulation. c Schematic diagram of key generation based on the randomly assigned polarity of antivortex core in a PUFs consisting of an array of $\epsilon\text{-Fe}_2\text{O}_3$ nanosheets.

Q3.2: *In addition, I cannot see the meaning of cross-tie walls as mentioned in conclusion. Can author elaborate this in the manuscript where they want to explain this cross-tie wall?*

Response: We appreciate the reviewer for pointing out this important question, which has prompted us to re-examine the relevant content and conclusions in the manuscript. After careful consideration, we agree with the reviewer's perspective that the discussion on the cross-tie walls in the current manuscript indeed lacks sufficient explanation and support, and its presence does not make a substantial contribution to the overall research. Therefore, we removed the mention of the cross-tie walls in the revised manuscript to ensure the accuracy and rigor of the content.

Q3.3: *In Fig 3a, I do not understand why the in-plane magnetization (black arrow) directions are same after the polarity changed; if the polarity changed, why not direction?*

Response: We thank the reviewer very much for pointing out this issue. There are two fundamental freedoms, namely polarity P and circulation c , which describe the antivortex. Since our samples lack a chiral magnetic interaction (such as the Dzyaloshinskii-Moriya interaction), P and c are not coupled together. Both our experiments and simulations reveal the existence of four types of antivortices with identical total energy, characterized by the combinations $(c, P) = (1, -1), (1, 1), (-1, -1),$ and $(-1, 1)$, as illustrated in **Figs. R9, R10**. Consequently, a change in polarity does not necessarily entail a change in chirality.

We have incorporated a discussion on chirality into the revised manuscript, see **Page 8**:
“There are two fundamental freedoms, namely polarity P and circulation c , which describe the antivortex. Since our samples lack a chiral magnetic interaction (such as the Dzyaloshinskii-Moriya interaction), P and c are not coupled together. Both our experiments and simulations reveal the existence of four types of antivortices, characterized by the combinations $(c, P) = (1, -1), (1, 1), (-1, -1),$ and $(-1, 1)$ with a topological charge $Q = 0.5P$ (Supplementary Fig. 10). Four styles of antivortices are degenerate with identical total free energy (Supplementary Fig. 11).”

Figure R9. (a) MFM phase images, (b) simulated magnetization distribution, and (c) simulated MFM phase images of four energetically degenerated magnetic antivortex states.

Figure R10. Four-fold degeneracy of antivortices in the truncated triangular multiferroic ϵ - Fe_2O_3 polycrystals. Simulated total energy of four styles of antivortices.

Q3.4: In another schematic figure authors should clearly indicates the specific thermal transition time (for example critical time etc.) when T_c change with time, gets plateau and again back to same.

Response: Following the reviewer's nice suggestions, we annotated the time (critical

time) when the antivortex is refreshed in the schematic diagram of **Fig. 3a (Fig. R8a)** to enhance the conveyed information. During the thermal annealing process, the magnetic antivortex first undergoes a heating process, where it is heated from room temperature (RT) to the annealing temperature (T_A) (above a magnetic transition temperature of $\epsilon\text{-Fe}_2\text{O}_3$). However, during the heating process in a tube furnace, temperature distribution may become uneven due to factors such as the thermal capacity and thermal conductivity of the workpiece or medium, as well as the structure of the equipment. It is necessary to maintain the temperature for a certain period of time (in our case, 5 minutes) to ensure a uniform temperature distribution within the furnace. At the end of the isothermal process, which corresponds to the nominal critical time, the magnetic moments inside the nanosheets regain order, evolving into a new magnetic antivortex.

***Q3.5:** In conclave shape schematic, what are the green and gray arrow representing? May be in-plane magnetization but it is good to clarify the reason of using two different colors.*

Response: The green and gray arrows in the schematic diagram indeed represent the in-plane magnetization. The use of green arrows is intended to emphasize the characteristic of the magnetic antivortex, where spins flow towards the core from two opposite directions and flow out of the core from two other opposite directions. Thank the reviewer for their meticulous observations and valuable suggestions. The relevant information in the manuscript has been clarified.

***Q4:** In addition to Supplementary Fig 6, can authors examine how the distortion affect in evolution of such antivortex? I would like to see the distortion of the antivortex structure (may be vortex creation followed by annihilation and form new anti-vortex) induced by an in-plane field? It might help to visualize and predict the new anti-vortices in such materials if distortions play the role in originating new antivortex. It would be great if authors can provide experimentally as well as magnetic simulation. If not experimentally because of limitation, I'm fine with magnetic simulation. For this, paper 10.1103/PhysRevB.77.060404 may be helpful.*

Response: Following the reviewer's nice suggestions, we have further studied the antivortex distortion induced by an in-plane field stimulation. Because of experimental limitations, we provide only simulations (**Fig. R11**). Setting antivortex as the initial

state, we apply a pulsed in-plane Gaussian-shaped field B_{xy} expressed by $B_{xy} = B_0 \exp(-\frac{(t-0.5 \text{ ns})^2}{w^2})$, and here B_0 is the peak field; t is the time, and w corresponds to the pulse duration. By varying B_0 and w , we obtain the dynamic process after the application of the in-plane pulsed field along the y -axis. When applying the field stimulation, the vortex could shift and disappear, as well as the formation and annihilation of new vortices. Interestingly, for approximate B_0 and w , we obtain new stable states, *i.e.* a vortex-antivortex pair.

We have added a discussion about antivortex dynamics triggered by in-plane magnetic fields in the revised manuscript, see **Page 11**: “*A previous theoretical study demonstrated the emergence and disappearance of new vortices during the stimulation of pulsed in-plane magnetic fields in patterned films (Phys. Rev. B 2008, 77, 060404). In this work, we delve deeper into the ultrafast magnetic dynamics of initial antivortices in truncated triangular multiferroic ϵ -Fe₂O₃ polycrystals through simulations (Supplementary Fig. 21, Supplementary Videos 1,2). We utilize a Gaussian-shaped pulsed in-plane field described by the equation $B_{xy} = B_0 \exp(-\frac{(t-0.5 \text{ ns})^2}{w^2})$, where B_0 represents the peak field; t is time, and w corresponds to the pulse duration. Our simulations reveal a rich array of vortex dynamics, including shifts in the initial core of the antivortex and the creation and annihilation of new antivortices or vortices. These topological defects commonly appear and dissipate at the geometrical edges and grain boundaries. When the pulse width w or peak field B_0 is small, the initial antivortex retains its configuration. However, for stronger or longer pulses, we observe a vortex-antivortex pair as the final configuration. With even stronger and longer pulses, triple-domain states emerge as the stable final configuration.*”

We have also added **Supplementary videos 1, 2** to show the dynamic transformation from initial antivortex to vortex-antivortex and triple-domain states.

Figure R11. Final stable phase triggered by in-plane pulsed field. (a) A stable phase diagram varying with the peak field B_0 and w . (b) Magnetic configuration of the vortex-antivortex pair.

Supplementary video 1 | Dynamic transformation from the initial antivortex to vortex-antivortex pair driven by in-plane pulsed fields. $B_0 = 1250$ Oe and $w = 24$ ps.

Supplementary video 2 | Dynamic transformation from the initial antivortex to triple-domain driven by in-plane pulsed fields. $B_0 = 1250$ Oe and $w = 36$ ps.

Q5: The reviewer is curious to see the magnetization properties (magnetization vs field and magnetization vs low temperature (5K-350 K). Usually at low temperature magnetocrystalline anisotropy (MCA) changes as compared to room temperature because of geometry change of Fe sites in the crystal structure. I wonder this stable antivortex still exist while changing the MCA.

Response: We thank the reviewer for this professional question. According to that, we made the following discussions.

Firstly, we carried out the vibrating sample magnetometer characterizations from a physical property measurement system. As shown in **Fig. R12**, temperature-dependent magnetization measurements were performed on the sample with the magnetic field parallel to the substrate (in-plane direction) during both the field-cooling and zero-field-cooling process under an external field of 500 Oe. The temperature-dependent magnetic features of 2D ϵ - Fe_2O_3 polycrystals are similar to those reported in previous research on ϵ - Fe_2O_3 nanoparticles (*Chem. Mater.* 2010, 22, 6483). The magnetization undergoes obvious changes at two notable temperatures, namely ~ 200 K and ~ 160 K. According to previous studies, these magnetization changes are because ϵ - Fe_2O_3 happens a second-

order structural transition presumably caused by changes in the coordination of Fe_A and Fe_D sites. These structural and magnetic transitions weaken the MCA, resulting in a decrease in the MCA constant (*Chem. Mater.* 2006, 18, 3889). In addition, under the parallel magnetic field, magnetic hysteresis loop can be clearly observed at 300 K (**Fig. R12b**), with a coercivity value of ~4 kOe.

Due to instrumental limitations, we conducted an *in-situ* temperature-variable MFM measurement in a limited temperature range (250 K-350 K). As shown in **Fig. R13**, the magnetic antivortex texture did not change during the measurement, and the magnetic antivortex core at the center of the sample can always be observed. Furthermore, we explored the stability of the antivortex through simulations. As per the referenced literature discussing MCA, it has been suggested that MCA decreases at low temperatures (*Chem. Mater.* 2006, 18, 3889; *Phys. Rev. B* 2009, 79, 094404). To demonstrate the magnetic evolution as MCA decreases from the initial antivortex and vortex states, we have included a simulation in **Figs. R14, R15**. Our simulations reveal that the antivortex can persist as a metastable phase even when MCA diminishes to zero, albeit with a higher total energy compared to the vortex.

We have added a statement on *in-situ* variable temperature MFM measurement into the revised manuscript, see **Page 11**: “Furthermore, *in-situ* temperature-variable MFM measurement indicates that the antivortex texture remains stable within the available temperature range (250 K-350 K) that the instrument can achieve (Supplementary Fig. 19).”

We also have added a discussion about the effect of MCA on the stability of antivortex in the revised manuscript, see **Page 9**: “The MCA is crucial in stabilizing the antivortex. Consequently, we also delve into how MCA impacts the stability of the antivortex by simulations. By starting with the antivortex as the initial state, we observe that although the magnetic configurations can change, they remain as antivortex even when the MCA drops to zero (Supplementary Fig. 15). For low MCA values, the antivortex can exist as a metastable phase, possessing a total energy that is higher than that of the vortex (Supplementary Fig. 16).”

Figure R12. (a) ZFC (blue line) and FC (red line) magnetization curves of the ϵ -Fe₂O₃ nanosheet with a magnetic field of 500 Oe parallel to the substrate. The arrow indicates the characteristic temperature at which the magnetic transitions occur. (b) Magnetization versus magnetic field curve at 300 K with the magnetic field parallel to the substrate.

Figure R13. (a) AFM image of ϵ -Fe₂O₃ nanosheet. (b-g) MFM phase images recorded at 250, 260, 280, 300, 325, and 350 K, respectively. (h) The corresponding model of magnetization circulation.

Figure R14. Simulated magnetic evolution by decreasing the magnetocrystalline anisotropy. (a) Magnetic evolution from an initial antivortex. (b) Magnetic evolution from an initial vortex.

Figure R15. Total energy as a function of magnetocrystalline anisotropy for antivortex and vortex.

Response to Reviewer 3:

General Comments:

Response: Thank the reviewer for his/her precious time on reviewing our manuscript. After carefully reading the detailed comments from you and other reviewers, I have given serious consideration to each of the specific points raised and made meticulous revisions to the current manuscript to improve its quality.

List of Responses and Revisions

Title: “Stable antivortices in multiferroic ε -Fe₂O₃ with the coalescence of misaligned grains”

Authors: Wuhong Xue, Tao Wang, Huali Yang, Huanhuan Zhang, Guohong Dai, Sheng Zhang, Ruilong Yang, Zhiyong Quan, Run-Wei Li, Jin Tang, Cheng Song & Xiaohong Xu

ALL THE CHANGES WERE DONE IN RED IN THE REVISED MANUSCRIPT.

Response to Reviewer 1:

Comments:

In this revised version, the authors have well addressed my previous concerns. I'm satisfied with their responses and revisions. I'd like to recommend the acceptance.

Response: Thank you very much for your kind and positive feedback on our revised manuscript. Your constructive suggestions have been invaluable in improving the quality of our work.

Response to Reviewer 2:

Comments:

I am satisfied with revised versions and authors response and recommend the current version of the manuscript for publication in Nature Communication.

Response: Thank you very much for your thorough review and positive recommendation of our revised manuscript, and we are grateful for your time, effort, and expertise in reviewing our work.

Response to Reviewer 3:

Comments:

Response: I would like to extend my heartfelt thanks to both of you for your meticulous co-review of our manuscript. We are thrilled that our manuscript has been accepted for publication in principle, and we owe a significant part of this success to your expert guidance.